# Psychosocial and Organizational Processes and Determinants of Health Care Workers’ (HCW) Health at Work in French Public EHPAD (Assisted Living Residences): A Qualitative Approach Using Grounded Theory

**DOI:** 10.3390/ijerph18147286

**Published:** 2021-07-07

**Authors:** Anne Armant, Florian Ollierou, Jules Gauvin, Christine Jeoffrion, Baptiste Cougot, Mathias Waelli, Leila Moret, Kristina Beauvivre, Ghozlane Fleury-Bahi, Gilles Berrut, Dominique Tripodi

**Affiliations:** 1Work and Health Innovation Research Laboratory, Department of Occupational Medicine and Environmental Health, Nantes University Hospital, F 44093 Nantes, France; annearmant@gmail.com (A.A.); florian.ollierou@chu-nantes.fr (F.O.); jules.gauvin@chu-nantes.fr (J.G.); Baptiste.COUGOT@chu-nantes.fr (B.C.); kristina.beauvivre@gmail.com (K.B.); 2Laboratoire de Psychologie des Pays de la Loire LPPL-EA 4638, Department of Psychology, University of Nantes, F 44000 Nantes, France; Ghozlane.Fleury@univ-nantes.fr; 3Laboratoire Inter-Universitaire de Psychologie, Personnalité, Cognition et Changement Social LIP/PC2S—EA 4145, Université of Grenoble Alpes, F 38058 Grenoble, France; christine.jeoffrion@univ-grenoble-alpes.fr; 4Qualité de Vie et Santé Psychologique (QualiPsy) EE1901, UFR Arts et Sciences Humaines, University of Tours, F 37000 Tours, France; 5Management des Organisation en Santé MOS, EA 7348, Ecole des Hautes Etudes en Santé Publique EHESP, F 35043 Rennes, France; mathias.waelli@ehesp.fr; 6Methods in Patients-Centered Outcomes and Health Research-SPHERE, UMR INSERM U 1246–EA 4275, 22 Bd Bénoni Goullin, University of Nantes, F 42200 Nantes, France; leila.moret@chu-nantes.fr; 7Gerontology Department, Pôle Hospitalo-Universitaire Gérontologie, Nantes University Hospital, F 44093 Nantes, France; gilles.berrut@chu-nantes.fr

**Keywords:** nursing home, social determinants of health, occupational health or health at work, burnout, demands and resources, social roles, qualitative approach, grounded theory

## Abstract

In a context marked by negative health indicators that make structural aspects more salient, this paper aimed at understanding and explaining the processes and determinants at work that positively and negatively interfere with the professionals’ health in the French public nursing home environment. To this purpose, the qualitative approach by grounded theory was chosen. In total, 90 semi-structured interviews were recorded and 43 were transcribed; in addition, 10 observations of 46 participations in meetings and working groups were carried out in four public service and hospital establishments. Our results indicate that the role of health workers, its definition, and its execution are fundamental to the understanding of their health at work. Two protective and constructive processes are involved in the maintenance and development of the professionals’ health in this work, with considerable confrontations with death and suffering: individual and collective control of emotional and cognitive commitment, and the development of resources for formation, information, and cooperation. Nonetheless, they are jeopardized when a lasting imbalance is generated between the work’s demands and the available resources. This leads to a loss spiral in organizational, inter-individual, and individual resources that makes it difficult to sustain work.

## 1. Introduction

The French system of nursing homes, EHPAD (Etablissements d’Hébergement pour Personnes Agées Dépendantes), was already going through a major crisis before the occurrence of the COVID-19 epidemic in the spring of 2020, with important repercussions on working conditions. The absenteeism rate (8.9% in EHPADs compared to 6.6% in 2015 for the health sector as a whole), the number of work accidents (almost twice the national average for all activities combined), the staff turnover, and recruitment difficulties (almost 10% of nursing auxiliary posts not filled in 2015) show the extent of the crisis [1,2]. Among the international variety of *nursing homes* [3], French EHPADs are accommodation and long-term care establishments. Within the various professions that operate there, the main one, responsible for dealing with dependency, is that of care assistants (nursing assistants, medical–psychological assistants, and gerontological care assistants), who account for 39% of the full-time equivalents (FTE) of all staff. The second staff category is that of agents (hospital and social service agents) who are in charge of accommodation and catering services. They represent 22% [4]. The rest of the staff is made up of nurses (11%), general services staff (11.8%), management and administrative staff (6%), psychologists (2.6%), physiotherapists and other paramedical staff (2%), educational, pedagogical, and social staff (2%) and, finally, medical staff represent 0.8% of FTE. Beyond the EHPADs, the problem of health at work has been intensely felt throughout the French public health service since management policies, introduced in the 1980s, reduced the resources of care professionals [5,6].

A study carried out between 2003 and 2005 in ten European countries with a representative sample of 39,898 paramedics (including 5376 French paramedics) shows that although France has a higher burnout rate than its neighbors (46% compared to 26.1% overall), it is in French retirement homes and geriatric services that the prevalence of disorders is the highest (50.1% in France compared to 32.5% overall) [7]. Such prevalence strongly warrants attention to mental health at work in these organizations, especially as this international comparison highlights that the Netherlands has a burnout rate nine times lower than France among paramedics [5]. This research might help to identify structural predictors of exhaustion in order to prevent it, in the same way research on magnet hospitals has identified predictors of staff satisfaction and turnover [8,9]. After or before individual diagnoses of burnout, organizations should be diagnosed on the basis of tangible organizational criteria in order to promote primary and secondary prevention. Beyond a simple assessment of the level of burnout risk, such an approach could provide institutions with concrete avenues for organizational transformation and thus facilitate their commitment to prevention.

Relationships between individuals, their mental health, and their mode of social inclusion have been analyzed for over a century [10]. Nowadays, although the scientific literature recognizes the impact of an individual’s social (including professional) integration on his or her health, this knowledge is not yet widely available [11]. More specifically, research conducted since the 1970s and 1980s on stress and burnout has identified several mechanisms linking work environment and health. Firstly, Karasek’s works [12] have emphasized the negative effect of important professional requirements when they are not associated with decision latitude (or control) and social support. During the next decade, Siegrist [13] showed that material and symbolic gratification (money, consideration, status control) made it possible to counteract the negative effect of the efforts made at work. To summarize, work organization can have a protective effect on the employees’ health by enabling effective adaptation, insofar as it promotes decision-making latitude and social support, and rewards the work done.

In the 1980s, Maslach and his fellow co-workers [14] emphasized the link between professional exposition to suffering and exhaustion. More specifically, they suggested that exposure to the suffering of patients is a job requirement, which is likely to consume the individual’s emotional resources and eventually lead to emotional burnout [15] in the long term. While the literature shows that emotional demands are positively associated with burnout, some works show that general stressors, such as work overload and role conflict, are more strongly correlated with emotional exhaustion than specific emotional demands, such as death or the severity of the client’s problems [16]. In other words, it is not only the exposure to suffering that is the problem, but also the way in which the organization gives (or does not give) teams the necessary resources to deal with it. Most recently, after Karasek’s [12] and Hobfoll’s works [17], the model of Job Demands–Resources (JDR) has had a resounding scientific impact [16,18]. The latter shows that while the demands of the job are a major factor in the prevalence of burnout, the resources provided by the organization, when they are perceived as sufficient, produce a commitment to work on the part of the professional, while, on the contrary, their insufficiency produces withdrawal. To summarize, in order to fully understand the links between work organization and health, it is necessary to pay as much attention to the demands as to the work’s resources. In the French scientific literature, the same movement can be observed: in the 2000s, studies on psychosocial risks (PSR), in other words, the social risk factors for psychological disorders, came to the fore. Askenazyet al.’s report, commissioned by the French government, summarizes a body of work and maps PSR into six categories [19]: work intensity and time; emotional demands; autonomy at work; social relations at work; ethical suffering; insecurity of the work situation. Some time later, Yves Clot [20] was invited to look at the psychological and social resources associated with good work rather than focusing on the risks.

In parallel with the work on demands and resources, interest in the social (including professional) integration of the individual has developed. The focus is on the relationship between the role that an individual occupies in a society through work, his or her resources to carry it out, the resulting meanings, and his or her health. Thus, for Cherniss [21], burnout is rooted in the work environment and is related to the job role. Burke [22] empirically tests the theory of identity and validates the hypothesis that stress is a consequence of an individual’s difficulties in fulfilling his social role. Other works will demonstrate the impact of professional identity [23] and social identity on burnout [24]. In the French literature, over the past decade, there has been a growing body of work on the relationship between an individual’s social integration through work and his/her health [25,26].

The wealth of work developed on the psychosocial dimension of mental health at work over the past half-century should be highlighted. Nevertheless, several limitations can be noted, the first of which is set out by Desgroseilleirs and Vornax [27] for whom the understanding of reality is hindered today by fragmentation due to the factor-based approaches that prevail in hypothetical-deductive methodologies. A second limit proposed by Schaufeli and Taris [28] about the JDR theory was its weakness in the application of mechanisms underlying the achievement of the results. This criticism could be extended to research that uses this type of quantitative methodology. The qualitative methodology that we propose using, anchored in field data through direct observations and interviews centered on the analysis of work units, allows us to get as close as possible to the activity carried out and helps to establish a link between the various elements of mental health at work. It allows multi-level observations: individual, inter-individual, and organizational. By paying attention to structural conditions, determinants, consequences, processes, and actors’ strategies, it offers the possibility to understand what is at stake in the real worlds that exist behind the models built with statistical methods [29].

Therefore, this article sets itself the following objective: to identify, understand, and explain, in a context marked by negative health indicators, the processes, and determinants at work that positively and negatively interfere with the professionals’ health.

## 2. Material and Method

### 2.1. Design

This study is part of an interventional research project aimed at improving the quality of life at work of professionals in EHPADs through the implementation of an empowerment program for care teams. In this project, two groups of retirement homes, an “intervention” group (1) and a “control” group (2), were composed from the diversity of retirement homes, making up the public service sector in France (Figure 1). Each group included, after randomization, two retirement homes from the Territorial Public Service (TPS) and two retirement homes from the Hospital Public Service (HPS), one of each being in a rural area. There was a total of 8 EHPADs whose management volunteered to participate in the study. The promoter of the research project was the University Hospital (UH) of Nantes, and the funder was the national pension fund of local authority staff. It was financed with €244,640 [30], which includes a budget allocation for the structures to cover the agent time taken for research.

For the present study, data collection and analysis were guided by the principles of grounded theory. It consists of a method of inductive analysis of data collected in the field, in this case through interviews and ethnographic observations, in order to arrive at a singular understanding of a given problem. We focused on the understanding of real work, its determinants, health states, and the interactions between the two among health care workers: health care aides (HCAs), medical–psychological assistants (MPAs), hospital service workers (HSWs) and territorial social workers (TSWs), and state-registered nurses (SNs). Three types of data were collected and analyzed: semi-structured interviews (see interview guide in Appendix A); ethnographic observations of work situations; and participation in working groups and meetings for the diagnosis and improvement of working conditions.

### 2.2. Setting and Population

The study population was selected in the following way: information was sent to all hospital and territorial EHPADs in one French department (Loire Atlantique). Of these, 8 EHPADs volunteered to participate. Then, a draw determined the establishments in the intervention group 1 (which concerns our study, see Table 1) and the establishments in the control group 2.

Then, all of the professionals in these establishments were informed by their management of the opportunity to participate in this study during their working hours, and all of the HCW volunteers were included.

Then, the following occupational interviewees were selected for this paper: health care aides (HCAs), medical–psychological assistants (MPAs), gerontological care assistants (GCAs), state-registered nurses (SNs), and agents in charge of accommodation and catering services (hospital and territorial service workers (HSWs) and (TSWs).

A total of 90 semi-structured interviews were conducted and recorded using a digital recorder for the research and project purposes. The interviews have an average duration of 34.96 min, with a deviation standard of 12 min. Of the 90 interviews undertaken, 43 were selected on the basis of occupation, establishment, variety of experience, and relationship to work (beginners, skilled, experts/exhausted, healthy/motivated, disengaged …) (see Table 2 for the description of the distribution of interviews), in order to achieve the required level of qualitative data, i.e., such that the other interviews did not provide any additional information.

Ten direct observations of half a day each, spread over the four intervention group establishments, were carried out during morning and lunchtime care periods. Each observation was accompanied with handwritten notes and photos and was then transcribed.

Participation in 46 meetings, gatherings, and working groups with the management, supervisors and teams of the four institutions whose reports and observation notes provided input for the analysis were performed. During these meetings and working groups, the participating observer had the status of ergonomist and sociologist assisting in the improvement of the quality of life at work and facilitating the implementation of the empowerment program.

### 2.3. Data Collection

The dataset was collected by Anne Armant from the 4 establishments between the end of 2018 and March 2020. The interviews all took place at the workplace, with each professional being given a slot to participate, with replacement, during their working time. The interviews took place between the end of 2018 and the beginning of 2019, the observations of the work situations took place in the spring of 2019, and the participation in meetings took place mainly in the period from January 2019 to March 2020.

### 2.4. Analysis

The analysis of observed data and interviews was undertaken as follows: 16 semi-structured interviews were selected from the 43 transcripts on the basis of the variety of cases and these were transcribed in full and then codified, which was summarized in units of meaning in the margin. The choice of this sample was based on knowledge of the content of the interviews, so as to provide a full variety of experiences and relationships at work as well as of the professions and establishments involved. The reflections of the professionals during the interviews concerning their work, working conditions, and health was also a factor influencing inclusion. The first hypotheses were drawn from these interviews and worked upon in the diversity of the cases represented. Then, a set of analytical operations was applied to the data to obtain a singular understanding of it [29,31,32]. For example, it is a question of identifying “concepts of practice”, of going up in generality by case, of looking for negative cases, of looking for a variety of phenomena and comparing them [29], of classifying the different elements into structural conditions, behaviors, representations, and effects.

The results of the 10 observation periods were introduced after this first conceptualization and helped to complete it. The remaining 27 semi-structured interviews, out of the initial 43 transcripts, were used for the back and forth work between the analysis and the data. Without constituting the basis of the analysis, as in other research [33], the “R interface for Multidimensional analysis of Texts and Questionnaires” (IRaMuTeQ) software (Pierre Ratinaud, LERASS, Toulouse, France) was used to facilitate the work of going back and forth between the data in the total corpus of 349 pages of the transcribed interviews. The hierarchical clustering method was used to check, compare, and complete the manual analyses. However, the first use of the software was the classic text statistical analysis, which allows the researcher not only to list all the words present in the corpus and their frequency of use but also to gather all of the sentences in which they are contained, such as for example for the word “time” used 1044 times or “resident” used 523 times.

The interviews with the managers were not included in the analysis of the verbatim reports because they were few in number and not representative of the perceptions of the healthcare workers themselves, who were the subject of the study. However, their words enriched the discussion so that their participation in working meetings and focus groups served to mutually enrich the reflections of the researchers and of the actors in the intervention. They also encouraged the professionals to express themselves on the analysis proposed by the constructed diagnosis, in order to enrich the reflection on their work and their health at work, which, in turn, increased the accuracy of the research’s reflection of the situation [34] (Figure 2).

### 2.5. Ethics

Each professional in the selected establishments was given, on a purely voluntary basis, the opportunity to participate in the study. Each received a letter explaining the study with its objectives, its confidentiality, and its anonymity framework. A letter of consent outlining the terms of voluntary participation and anonymity was signed by each participant. The protocol was validated by the ethics committee; its reference number is RC18_0089-QVT. The workplace health and safety committees (WPHSC), in accordance with the French labor law, approved the study protocol.

## 3. Results

The interview participation rate is 43.6% on average, ranging from 15.1% to 76.3% (Table 3).

### 3.1. Forms of Engagement and Self-Realization in a Role of Complex and Protean Support

In total, 92% of the professionals we met were women, mainly from the working class (blue-collar workers, employees, craftsmen), who had sometimes chosen their occupation from childhood, having learned about it through their family environment. For the agents and medical care assistants, the interest to work in gerontology is often born from a professional internship. It has also often been the result of a reorientation process following an initial employment in an industrial environment and a decision to reorient to an activity more aligned with their values. Finally, in a period where there is a lack of personnel coupled with a high unemployment rate, some people first turned to these trades out of opportunity in order to access employment and income. In general, what inspires them is the “human” involvement: taking care of and being of service to people in need.

In addition, it appears from the interviews that the “own role” that the health care workers give themselves is often more extensive than that defined by their official task sheet. This “own role” is defined by the professionals primarily by dispositions toward work and residents: love, patience, kindness, and empathy (Table 4). These *dispositions* are associated with four *objectives* that guide the action of professionals toward residents:(1)To develop the desire and the joy of living,(2)To ensure the comfort and cleanliness of residents and their housing,(3)To participate in the maintenance of the aesthetics, and(4)To protect the physiological health of residents.

Similar objectives have been observed in the United States [35].

To implement these four objectives, we find four guidelines defined by the professionals:(A)Letting go, to let the residents perform all the actions they can still do themselves,(B)Let the resident people decide for themselves, in order to maintain in them the feeling of power (in the sense of capacity) and the feeling of control over their lives.(C)The search for stimulation to avoid “slackness”, and(D)A personalized attention.

These guidelines are updated during the implementation of support centered on the resident’s individual needs and organized around three axes of support:(a)The conative, which consists of animating life, maintaining the desire and the momentum of life in order to avoid solitude, immobility, and sadness;(b)The cognitive, which supports a way of realization and avoiding confusion and disorientation. It is to help the residents to find their way; and(c)The physiological in the assistance of weakened bodies, which appears as the most prominent object to the uninformed observer by the diversity of the work of nurses, without letting appear the needed strong expertise for these very delicate actions.

Eventually, a diachronic and relational dimension is deployed in this role through personalized attention and the projection of the support for the journey of the resident in the establishment.

Those aspects of the role allow us to design a conceptual map of health caregiver’s engagement and self-realization (Figure 3).

If this role is very complex for professionals, it is mainly because the distribution of resources in terms of time, information, and training is not sufficient to fulfill the expectations of the role. Furthermore, the care of certain residents, in the absence of an explicit request and/or in the presence of serious pathologies, requires unusual knowledge and know-how.

Finally, professionals are regularly caught between two logics, those of *cure* and *care*, which contradict each other in certain situations and/or ways of reading them. For example, the residents’ wishes do not necessarily correspond to the clean and physiological care that the careers wish to provide. In this case, professionals will find tactics to obtain the resident’s consent. For example, it is a question of explaining the possible consequences of the absence of care, or even the exploration of the possibilities opened up by the latter. If this does not work, as may be the case with highly cognitively impaired people who would prefer to remain soiled, a non-consensual procedure may be conducted on the resident’s body. These “care refusals” are difficult both for the careers, who are frequently hit, spat at, and/or insulted, and who find themselves in contradiction with their objective of bringing joy, and for the residents as well. To deal with these constraints in carrying out their role, these careers resort to two forms of individual and collective protection: a control of self-commitment and resource development.

### 3.2. Individual and Collective Control of Commitment: A Condition for a Sustainable Commitment

In nursing homes, suffering and death are part of the landscape, and part of the work of all professionals is characterized by emotional support for both residents and families. To do this, everyone makes a point of trying to leave their problems at the door of the establishment and to present themselves “healthy” and relaxed. “When we are not feeling well, the residents feel it right away” (Mrs. T., AS, FPT (French Territorial Public Service)).

Within this framework, a careful balance is gradually established in the emotional relationship with the residents. Many professionals explain that “attachment” to residents is imponderable, although it is necessary to protect oneself from emotional invasion to be effective. They explain that they put “barriers” between themselves and the residents and work to protect themselves. They manifest themselves as the gap between home and work. Voluntary work on their thought flow is done on the journey: “I force myself not to think about it” (Mrs. A., AST, FPT). This work also takes place between the different tasks and residents with whom the professionals intervene. It is a question of being in the present, of leaving the pressure of collective organization at the door, in order to meet the needs of each person: “when I am with them, I am only with them” (Mrs. F., AS, FPH). This emotional and cognitive “distance” of all the expectations that weigh on the professionals is not simple and requires a certain amount of assurance. “Young, we want to do the work properly, we want to do it for the task sheet that we have to follow, we have to step back from it, but it’s complicated” (Mrs. F., AS, FPH).

Sometimes porous, protective barriers have disadvantages when they become too rigid. Indeed, the absence of an expression of personal difficulties results in incomprehension between colleagues and a lack of collective adjustment. Major difficulties with a sick child or a miscarriage in the toilet at work are kept quiet despite the deleterious effects they have on people. If in these cases the collective is inoperative, part of the work on modulating self-commitment is done in interactions with colleagues. Being able to exchange with colleagues, being supported in difficult situations, “handing over” and “letting go” are all mentioned in turn as important elements in coping with difficulties. These cooperations and their effectiveness are related to the understanding between colleagues, which is itself linked to the more or less strong sharing of points of view and ways of producing quality work. “A day with colleagues with whom we get along well, that’s everything, in the evening, we don’t have pain everywhere” (Mrs. C., AS, FPH).

Individual work is carried out by some with regard to the personal expectations of their colleagues in their way of working to avoid frustration and annoyance. Some managers participate in this work by guiding the mutual understanding of plural forms of doing and thinking. This type of work also extends to self-integrated expectations on the deployment of one’s activity, when significant resistance is brought into the work situations, either with the means at hand or with the expressed needs of the residents. “At the moment, I am working on accepting that there are days when I don’t go to the toilet [when the resident doesn’t want to], whereas some time ago, I wouldn’t have [said that I didn’t have a bad day at work], not doing a toilet, it would have been complicated” (Mrs. G., ASG, FPT).

### 3.3. Self-Development, Collective Activity, and Access to Information Resources

In nursing homes, “no two days are alike”, and in the variability of the work situations encountered, due in particular to the variety of pathologies and the variability of the residents’ condition, professionals, even experienced ones, are constantly learning. They learn from the gap between expectations, intentions, interactions, and the results of their actions, as well as from the observations and remarks of their more experienced colleagues and residents. The individualized knowledge of the latter is referred to as essential. In this process valued by development professionals, it was noted that the lack of time for reflection (individual or collective) in relation to practice leads to the adoption of automatisms and hinders the development of knowledge and skills.

Faced with the complexity of work and the more questionable automatisms, professional training appears to provide structure for activity and identities by reducing the level of doubt, increasing legitimacy, and providing tools to orientate one’s action. This is particularly the case for the training of a Gerontological Care Assistant (GCA), which is reserved for nursing assistants and medical–psychological assistants. In addition to a title, it provides theoretical tools adapted to cognitive disorders. With the right resources, the “puzzle” becomes an interesting “challenge”. Not everyone benefits from academic training in gerontology (GNA, end of life, Alzheimer’s, etc.), especially HSAs and TSAs, whose care role is seldom taken into account by the institution. Needs are expressed in interviews by professionals, particularly concerning cognitive disorders, the management of violence, or communication with families. Some of this knowledge is transmitted between professionals via “flash” training, break times, hallway work, “handing over”, or in collectivity situations. More specific training (massages, beauty care) allows them to engage in complementary ways of taking care of the elderly. This is particularly satisfying for the professionals who engage in it, in view of the satisfaction of the residents. 

While differences in initial and ongoing training impact the quality of professional links because of the resulting variations in work engagement styles, the opportunities for collaboration between professions are highlighted by several professionals as moments they appreciate. In this respect, some also mention their contribution in terms of task enrichment, knowledge, and ‘savoir-faire’ sharing, as well as a real knowledge of the constraints of each profession. On the contrary, the lack of communication between the occupations is the breeding ground for negative judgments about each other.

At the same time, daily and weekly situational information, particularly on the condition of residents, plays an important role as a necessary asset for quality work. “If I don’t know about it, how do I do it?” (Ms. E, SAT, FTP). Circulation of this information is partially organized, on written supports and by times of daily and weekly meeting: the “transmissions”. The adaptation of the support to the needs at work varies according to the establishment, and it is the same concerning the integration of the agents in “the transmissions”, meetings and access to the written supports, digital or paper, on the state of the residents. Other information circulates informally in the corridors and during break times. These times are also used to collectively solve problems encountered individually during the activity.

### 3.4. In Sustainable Dizziness between Expectations and External Resources: The Spiral of Loss of Resources and Control

These two processes, one protective (commitment control) and the other constructive (resource development) as realized in our observations, are challenged by the lack of time created by the mismatch between resource allocations and role expectations. Indeed, the two main difficulties reported in EHPADs are the frustration due to the lack of time to do the work and “the race” to try to do it all the same. This reported lack of time seems to be a series of consequences that impact the health of individuals and groups.

The direct attacks on physical health are played out in the lack of attention paid to the protection of one’s health due to a lack of time and/or, voluntarily, to gain it. For example, in their gestures, NAs, MPAs, and GNAs mobilize residents while preserving tired and fragile skin. Most of the day, they push, roll, support, and carry people who have difficulty moving on their own. They bend down and get up to wash feet, wipe them, put on socks, shoes, and tie laces. Handling equipment and practices support professionals in performing these physical tasks in a safe manner. However, when time is short, they are regularly sacrificed, the material is not used or is partially used, and the fastest postures may be preferred to the safest. “We have to hold on physically, we have to preserve ourselves, we have to use the means we are given. But the problem with using them is that it takes time that sometimes we don’t have” (Mrs. M., AS, FPH).

As for the psychological damage, it begins with the feeling of not providing adequate or sufficient answers to the residents. Especially since this feeling develops despite a multiplied intensity of effort. “It’s like that, there aren’t enough of us, they [the residents] pay psychologically, physically, and it’s always them who pay. Financially, well, in any way, they pay for care that is not received” (Mrs. M., AS, FPH). Sentences meaning that they do not work with or on objects and that the requested cadences are not adapted to the needs of the people come up regularly: “We do not work on cans”, “It is not vegetables that we wash.” The guidelines of “letting it be”, “letting decide”, and stimulation are limited, and so is the axis of psychological support. Mrs. H. (AS, FPH) says that she no longer wants to work in EHPADs; she is particularly affected by the lack of respect for the rhythm of the residents’ mealtimes in her establishment: “They are fed.” The meaning of work, in the way an individual does it, is impacted in this way: “I didn’t choose to do this job this way.” Pride, which refers to the self-esteem that comes from a job well done, is diminished. In addition, professionals appear alternately saddened or angry about the shortcomings of the support due to this lack of time. Mrs. P. (ASH, PFH) wants to leave and do temporary work for a while; she enjoys working with the elderly very much but can no longer bear witnessing the lack of activities offered to residents in her establishment. At the same time, within the tension to achieve these objectives, conflicts of views on the necessary prioritization, on the complex balance between cure and care, are multiplying, all the more when the professionals do not have the same frames of reference following training or years of training. Conflicts also arise over the balance between protecting oneself and providing commitment. In addition to this, absenteeism and high turnover, and limited communication and coactivity time, make it difficult to establish a common culture. Then, the tension mounts both inside and among professionals to try to get there without forgetting either your values, your health, or your life outside of work.

When the clock starts ticking for the day, to make sure the work is done on time, breaks are eliminated by some. The recovery rhythms are broken, and then, the professionals draw on their physiological resources. Collective adjustments made during breaks are removed. Thus, the resources of the collective are amputated. At the same time, sleep disorders are mentioned with recurrence, which certainly results from alternating hours but also from work-related concerns. When professionals are pressed for time, while they are moving quickly, bells ring out, “Please, please, please, please” ring out. It is impossible to respond to requests immediately; otherwise, the work would be nothing more than running back and forth between rooms. The “barriers” that the caregivers try to erect can fall when the difficulties in reaching the objectives increase: not being able to meet the expectations of the residents, looking for ways to get there in spite of everything, and having a disagreement with a colleague are all concerns that go beyond the framework of work, sometimes until late in the evening and at night. These cogitations also hamper resource recovery.

To cope with the physical and psychological intensification of work, professionals who do not have family responsibilities transform their leisure time into sporting activity “to build your back”, “to cope”, “to decompress”. Some work on a voluntary basis, arrive before the hour, leave after the hour, come to work sick; others take up family time, including during exceptional leaves of family life such as death. Schedules and work on weekends make it difficult to share time with family and friends. In addition, in some establishments that practice self-replacement to deal with absenteeism, certain days of rest are “skipped”. This disrupts private life and makes it difficult to distance oneself from work. In the case of childcare, some feel as if they are being left behind both privately and professionally. What is taken from private life is not only time but also the availability of the individuals’ internal resources which are at stake: “When I return home, I am empty, I have no more energy.” Parental identity is damaged along with professional identity. Mothers express in interviews their sadness at not being more present with their children. To maintain work–life balance, part-time work is often chosen when the second child arrives. Part-time ends up being kept in order to be able to maintain the capacity for modulating self-commitment. “It allows me to keep a good overview.” However, these choices are not allowed to certain professionals, who nevertheless would like them, for financial questions; in addition, some organizations are limiting their part-time offerings. Others, in order to save themselves from weekends and days off, prefer to remain in precarious contracts, which nevertheless allow them to accept, or not, to work on certain days. However, these choices keep them insecure and come at a cost to the organization as well.

This accumulation of difficulties (long-lasting pursuit of results, difficult recovery, and failure to meet objectives) creates the conditions for the loss of reflective control over self-commitment. In other words, there is a suppression of reflective activity during the action. The most experienced fear this phase, which they have identified and named “the head of the handlebars” or “the task mode”: a mode of operation which, by cognitive fatigue and the desire/need to go fast, is essentially based on automatisms. Mrs. G says that one day in her career, she told a resident who wanted to go to the bathroom: “We don’t have time, you’ll have to go in your pad.” A little voice lit up in her head without her having time to respond: “What am I doing?” It was time to settle the residents in for lunch, a particularly tense moment. After her shift, she walked into the locker room and broke down in tears looking at herself in the mirror and thinking “never again”. Some experienced supervisors pay attention to these states, which can be associated with negative thoughts. When they identify them, they suggest people take a rest, a vacation. They know that this is a risk area for the person and anticipate by proposing regulation at the individual level.

Indeed, then, there is only one step left toward the loss of self-image, as was the case with Madame G. looking at herself in the mirror, the individual no longer recognizes himself in his actions. Or Mrs. S., who says that it was her child who got her out of this spiral of resource depletion, when she had yet another reaction of annoyance against her: “Mom, you are always tired.” The spiral of loss of resources seems to end with a trigger that suddenly plunges the individual into disarray. This trigger, perhaps a highly significant error or instead of encouragement, recognition of effort, or suggestion of rest, a new addition of the load, the suggestion of multiplication of the effort, or globally of remarks perceived very negatively on one’s performance. The professional is not proud of the work he does and is even ashamed of the defects in the quality of the service provided. He is multiplying efforts to try to counterbalance the institution’s lack of resources; in doing so, it decreases its internal resources, especially as the recovery processes are also hampered.

In the context of a lack of material resources, feedback from the environment to validate the value of each professional then becomes all the more important. For instance, some professionals can benefit from more rewarding feedback from families they met at work, and it makes them look healthier and more relaxed than their co-workers. However, during this period, the social problem of maltreatment accentuates the critical view of the care provided in an assisted living residence as well as the lack of recognition of the investment and professionalism of the staff. This “professionalism” nevertheless binds them to continue to carry a load that cannot be endured over time.

### 3.5. The Dynamics of Loss of Recognition of Commitment 

In a view that often appears exaggerated and caricatured for professionals, the establishment of “mistreatment” as a public problem, in particular through the broadcasting of videos and television reports, has revived French society’s interest in work in EHPADs. In this advertising, which was deemed to be misleading, some find satisfaction in the form of recognition of the constraints they face in their activity, while others see it as a certain contempt for their work. Since the media boom, many have been asked by those around them about their profession or rather of the poor quality of support for residents and working conditions. As much as they are not understood, they managed to come out of a form of social indifference. If many of them explain that they suffer from a lack of recognition, it is because outside their professional sphere, a very narrow vision of their work is widespread. Ms. C, a young nursing assistant, cries in an interview when she says that some of her acquaintances summarize her job as “changing old people’s diapers”. In their own institution, their expertise and the breadth of their roles are often under-valued. Set up by certain directors and supervisors, practices about the acknowledgement of constraints and needed skills by observing the activity, taking “a trainee’s coat”, and listening attentively to professionals, manage to attenuate this feeling of lack of recognition. They are associated, in certain situations, with an enhancement of expertise and people, by colleagues and management, which likewise reassures people of their professional identity.

“Abuse” does not only exist on the screens and pages of the media, it also haunts the spirits in the corridors of nursing homes. The fear of doing wrong is a real issue, especially for novices, and with it, the fear of judgments from colleagues, superiors, and families. Conversely, the fear that others will do wrong is equally important. Although a category of mistreatment known as “institutional” has emerged, which makes it possible to question the environment and working conditions as the cause of mistreatment, it remains difficult to get out of an individual attribution of responsibilities in the face of this problem. Indeed, when the determinants of problematic work situations (resources available: time, training, equipment, information, support, organization) are erased to leave room only for an explanation in terms of intra-individual qualities (“skills”, “will”, and “motivation”) without understanding the links between the two, the discomfort is increased tenfold. Pairing undergraduates and novices with experienced supervisors reduces these fears. On the other hand, pressed by this question of mistreatment, by the “complaints” of families, the arrival of a new management, which seeks to improve the care, can, without knowledge of the causes, quickly devalue the feeling of justice and the personal esteem of the staff. It can also modify balances found somehow in the work to meet complex and sometimes contradictory requirements.

Negative social judgments—however inconsistent with reality—which are very difficult to shake off, feed feelings of injustice and guilt. Especially since, whatever the causes of the difficulties of the support provided, the failure remains difficult to accept for staff because of its effects on residents. Certain facilities, in situations of under-staffing, implement protocols to bear responsibility for this lack of resources. Thus, the responsibilities and the necessary priorities are not an accumulation of individual arbitration, which leads to guilt and to the lapidary judgments of families and colleagues as well as to a perpetual race against time. This does not prevent some professionals from doing more than what is requested, focusing more on the needs of residents than on the degraded mode protocols in place.

Finally, this media assault upon EHPADs is combining with a vocation and recruitment crisis, which is itself a major source of destabilization of the activity in the already complicated context of the workforce due to turnover and absenteeism. In this way, this dynamic of public communication arenas (media, and so on) and the social representations linked to the work in assisted living facilities intensifies spirals of internal resource loss for people and facilities by affecting staff resources and by associating negative symbols with this profession.

### 3.6. Workload Indicators Lagged behind Variations in the Condition of Residents and Unrelated to the Adaptability of Premises and Equipment

Beyond the under-staffing that all the staff talk about, which is associated with a minimum demand for the attendance of the planned workforce, professionals suggest a plurality of causes to explain the work overload. The first is the adequacy between the variability of residents’ conditions and the access to organizational resources (time, premises, materials, staffing), which may or may not allow the needs to be met in an appropriate manner.

The workload in assisted living facilities varies according to the “conditions” of residents. “How is he doing today?” This question is asked before accompanying each resident and is discussed in the corridors and during transmissions. The “conditions” of individuals with cognitive and psychological impairments are of additional concern. Regularly, events such as falls, illnesses, depressive episodes, temper tantrums, and paranoia add to the workload. Then, there are epidemics, influenza, and gastroenteritis, which, limited in time, significantly increase the workload. Faced with these circumstantial variations, there are sequential variations in the condition of the residents, for example, the loss of the “upright position”, the arrival of a swallowing problem. Each of these problems results in a substantial lengthening of the accompaniment time.

The workload also depends on the adaptation of premises and equipment to the activity. Proximity of intervention, meeting and rest areas, travel time, availability and proximity to the people being accompanied and colleagues, in order to be “in the right place at the right time” for handover, transform the work. The organization of the space is also important for the “corridor work”, where it is a question of monitoring the condition of the accompanied persons, being there and being able to be of service quickly, thus preventing crisis and accidents; welcoming and supporting families and outside workers; making regular updates on progress with colleagues and co-organizing and cooperating in carrying out and progressing the work. The size and organization of bedrooms, bathrooms, and furniture constrain or support work activity depending on the configuration. For instance, in some rooms, chairs and tablets have to be moved several times to allow residents to get up, wash, and clean. In some bathrooms, the space is so cramped that the entire toilet is done in a constrained position. The organization of places of sociability more or less accompanies the work to fight against the isolation of people. In some establishments, small tea rooms with game tables allow and invite discussions and group activities. In others, the alignment of chairs and armchairs along an entrance or a corridor gives an impression of waiting, the games and meeting rooms are the same as those for meals, and their availability is less and requires systematic rearrangements.

To these questions of space, furniture, and equipment, we must add socio-technical systems of communication between professionals and with residents, which are sometimes very ineffective and disrupt work more than they support it. In one morning, 43 calls were counted in the space of 2 h and 20 min on the telephone of a nursing assistant already in a hurry to provide care: an average of one call every 3 min.

The availability and working conditions of handling and handling aids, hammocks, stand-ups, trolleys, and electric beds alleviate the workload of officers. “It almost counts as one more person.” However, this availability and working order are not always guaranteed. Trolleys too high or too heavy result in movements that highly solicit the shoulders and make it difficult to push them, which can cause musculoskeletal health problems. Broken-down electric beds force workers to adopt risky positions. Defective batteries lead to a race between floors. Finally, when the resources are available and in working order, the necessary training to use them is not always guaranteed.

These elements, which determine the quantity of instrumental work demands, are not necessarily taken into account in the calculation of the workforce. In the assisted living facilities, the equation of the workforce and workload is done over sequences much longer than the actual variations in the quantity of work, which leads to very demanding periods for the personnel. By exploiting the limited room for maneuver, as the staff is recruited on three different budgets (Pathos medical care, GIR dependence, and accommodation), in accordance with standardized calculation grids, backup is nevertheless put in place in some establishments when peaks in activity are observed. Regulations are also made on the level of dependency of new residents in an effort to balance resources.

### 3.7. Skills and Organization Are Jeopardized by Absenteeism and the Management System

Absenteeism and its management is another determinant of work overload that comes up with insistence. One management board reports that the majority of its incumbents are absent because of long-term illness and ordinary illness. A study of the structure of absenteeism, combined with interviews with professionals, reveals that days of absence are concentrated among certain professionals, while others are not absent for a single day. The flip side of absenteeism is presenteeism. “We all have our own little grief”, “Here, we are all on antidepressants.” When the pain makes you double over or that psychological disorders suddenly paralyze you, out of solidarity for colleagues and for residents’ sake, sick leave is always used as a last resort. Worn out by the vicious spiral of resource loss, the sick leaves are then long.

From a management perspective, two types of absence, the long and the short unexpected ones, have different consequences. They are managed by different methods depending on the establishments: self-replacement, under-staffing, management unit of replacements for short contracts, and teams of substitutes in post. Some establishments, via an absenteeism management unit, have set the goal to replace each absence; yet the difficulties in carrying out the work are increasing. In situations when trained and experimented persons and with a good knowledge of the residents end up exhausted, in burnout, it is even more difficult for the inexperienced ones sometimes without training, without localized knowledge. The incumbents remaining on the job handle some of the workload of the replacements while passing on necessary information to the new employees for the proper execution of the work. Moreover, with referents by sectors, they will be responsible for the organization, information management, and availability of equipment, protections, sheets, etc. These are all tasks that novices cannot perform due to their lack of organizational knowledge. Others facilities mainly operate through self-replacement and under-staffing, leaving “lines” of incumbents empty for many months. Finally, there is the management of substitute teams on duty, in insufficient numbers compared to the absences, which is accompanied by self-replacement and under-staffing. The management of absences poses a constant challenge to the supervisory staff: “It takes me 70% of my time.” Although the organization of the “empty lines” planned can be anticipated, the problem of unexpected leaves remains. Supervision is provided by people from nursing functions, who are familiar with the constraints of the care professions. Marked with the ethics of care, they find themselves torn to ensure a minimum service by recalling staff at rest while trying to ensure a certain equality in the distribution of work. The succession of absences is a major fear of supervisors and when it begins; it gives rise to a series of undesirable events whose management in an emergency is described as “hell”, which takes up the time allocated to work organization of work and team support. Here, there is also a loss of organizational control, with emergency management taking the place of everything else. In addition, the turnover on these management positions is no exception to the rule.

### 3.8. “Levels” of Empowerment and Responsibility in the Workplace: Professional and Organizational Development Jammed by a Structure That Is Unsuited to the Deployment and Development of Skills

Another determining dimension of the multiplication of the workload and the degradation of the meaning of work lies in the boundaries that exist between professions. In the organization of work, specialization by status is associated with regulatory authorizations that sometimes prevent the action of professionals and the recognition of knowledge developed through experience and transmission between peers and professions. Thus, experienced nursing aides, who may have developed skills in monitoring the skin condition of residents and assessing the need for appropriate care (application of creams, hydrocolloid dressings, etc.), are regularly required to wait for the approval of the referring nurse even though their expertise would allow them to provide the necessary care. This lack of empowerment gives them the feeling of a lack of recognition and a lack of quality in care, which is caused by the period of authorization requests. This is all the more the case for a significant number of workers who have obtained a nursing diploma abroad and who, in the absence of validation of this diploma, hold the status of a care assistant in France. Some agents, for their part, have, for the most experienced, exercised the functions of a care assistant, “acting”, but since a rigidification of the legislation and practices, their access to these missions has been forbidden. They do not just have little access to training in the psychological and cognitive support functions they fill with residents, but they have also lost the ability to carry out functions for which they had developed knowledge through experience and the transmission of knowledge among peers. These regulatory barriers to the development of individuals and professional roles complicate the organization of quality work and its corollary, the quality of working life for professionals in establishments. In addition to the obstacles of real regulatory authorizations, there are also those of supposed authorizations, which sometimes, due to a lack of regulatory knowledge, increase the feeling of being unable to deploy oneself to offer quality care.

Furthermore, in the scale of material and symbolic recognition of occupations (each with different materials and symbolic rewards as well as powers of action and influence), there are difficult projections from one status to another, which is partly due, as mentioned above, to strong constraints in each occupation, which are perceived as repulsive. However, the lack of an individual and/or collective vision for change in problematic situations creates despair. A majority of ASTs and ASHs, who occupied some of the functions of “acting” care assistants before the rigidification of practices and regulations, do not now project themselves into the status of a care assistant, in particular because of the very high instrumental demand for physical and cognitive work in this profession. However, some experienced employees say they suffer from monotony in their work. Many care assistants do not see themselves in the current role of the nurse because the administrative workload and the management of the distribution of medicines keep them away from a relational work of proximity, in which the objectives of care for the residents are achieved. On the other hand, few nurses see themselves as managers because of the time they spend dealing with emergencies due to absenteeism. Moreover, the top of the scale of professions, held by doctors, is so difficult to reach in the French system that no thought is expressed regarding the transition to this status. Their time on the wards is also low, which separates them from the nursing staff and complicates the work of the nurses.

In addition to status-based authorizations, organizational decision-making authorizations are governed by management ‘levels’. In a particularly tense context, organization is an important activity and concern of professionals, supervisors, local and institutional management. What to share between the morning team and the afternoon team? Where to put intermediate stocks of protection? What kind of support should be put in place for Mr. Pidul? What day to clean the fridge? How to transmit information efficiently? When a problem is encountered in the activity, the search for individual and/or collective solutions takes place at work. Work and organizational concerns play a significant part in the activity of professionals. Some of them spend evenings volunteering and completing care sheets for residents in order to facilitate the work of novices and the quality of their work. A meeting in the corridor is an opportunity to discuss the synchronization of protection changes between night and morning shifts, to limit the waiting time of residents and optimize the intervention time, or the search for optimization of supply time by the localization of intermediate stocks of certain products. However, the organization of working resources is subject to a hierarchy of budgetary and regulatory rules, which is coupled with a decision-making hierarchy of varying degrees depending on the size of the organization and its operating rules. This leaves little power to solve problems, not only for professionals but also for the institutions, which then come up against recurring organizational problems because they are unable to intervene in their determinants (the persistence of the problem of managing absenteeism, the changes that have taken place, particularly in terms of staff turnover, with the difficulties of monitoring that this creates in the teams, as well as the changes in management and leadership, etc.). Just as some of the objectives that professionals set for themselves are not taken into account in the organization of the external resources provided by the organization, the organizational problem-solving work itself may have little place. Some note that when problems arise in their activity, the organizational procedure is to fill in a form, which they barely have time to complete, and for which they have no feedback. Without organizational support to intervene on the determinants of problems, organizational work can be less effective. As a result, a feeling of lack of control and despair sets in among some agents and nursing assistants, who were previously entrepreneurs of change with long experience in the institution. These feelings will themselves become disruptive elements in the already complex dynamics of improving the organization of work, leading to predictions of failure of proposals for improvement made by others, and in particular those made by new entrepreneurs of change who have recently arrived in the institution. Accompanied by complaints and the search for culprits or at least for people responsible—if no solution is possible—these feelings may in turn become the determinants of a status quo or regressive retroactive loop in challenging situations. The interpretations of these attitudes will vary according to institutions and individuals: from circumstantial and external causal attribution to the individual’s identity (fatigue, “burnout”, etc.) to those more internal under the register of personality (“negative”, “toxic”). Therefore, explanatory resources will also have their role to play in preserving social ties and identities.

At the same time, the maintenance of the problems reported in the achievement of objectives is perceived as a lack of recognition of efforts and experiential expertise, which contributes to the creation of grievances and mistrust toward institutional management. The latter, depending on the extent of their decision-making sphere, in addition to reduced regulatory and budgetary room for maneuver, find themselves with overloaded timetables, which limit their reactivity and power of action.

## 4. Discussion and Implications of the Research Results for Work Design in Assisted Living Residences

The major contributions of our research are as follows: First, there is the identification of eight sub-themes grouped into three “virtuous circles” dependent on each other and which can help improve the quality of life at work of HCWs and also for the elderly/residents/patients. A first “virtuous circle” includes five themes related to the HCWs: (1) Commitment, self-realization; (2) The link with the residents (“the right distancing”), (3) The access to resources/to information/to training/to good communication between all the professions. (4) Matching time with available resources, and (5) Fighting against effort/reward imbalances. A second “virtuous circle” corresponds to the search for good workload indicators: The number of treatments delivered? The degree of patient dependence? The type of work space allocated? The promise and use of ergonomic aids? The number of reported adverse events? The number of declared or non-declared task interruptions? A “third virtuous circle” concerns management with two major priority themes: the management of absenteeism with the question of improving the attractiveness of our establishments, and the attribution of levels of empowerment or skills, which are shared by all. Secondly, our first results have theoretical and practical implications. To our knowledge, this is the first “research-intervention” study assessing the impact of empowering work organization on assisted living residences (EHPAD) in France. These first results were presented to the different departments of the participating hospitals and then to the HCWs in order to implement local prevention plans. From a holistic perspective, this article contributes to the thinking about the structural and interactional dimensions of health at work. It attempts to consider the links between the evolution of individual health at work and the structuring of work’s health with its classic triad of objectives, resources, and results, to which we add the interpretation of these results. While research through hypothetical deductive and verification factors is multiplying and providing valuable evidence, it has the interest of providing, by returning, through a qualitative approach, to the real professional world and recreating the connection in the approach to the determinants of health at work.

### 4.1. Self-Realization through the Development of a Specific Role

This study highlighted the importance of the caregiver engagement and self-actualization through the acquisition and fulfillment of a role specific to the care of dependent older adults. That is to say, the development of a role that goes beyond the task sheet and deploys thought dispositions toward the work, the objectives, and the guidelines is important. Behind the concept of “well-done work” [20] or “good care” in the care sectors [35,36], we find how the work and its objectives are redefined by the workers and the essential place of the satisfaction they will feel regarding the work performed. Thus, understanding the role of support as it is defined in the activity and mindset of caregivers in assisted living residences is of utmost importance. Among the central categories observed in the magnet hospitals, there is the one of excellence in work, which from our point of view is linked to this concept of self-realization through the role [37]. Moreover, by conceptualizing this role, we can also observe similarities between the objectives and the values promoted by Hans Becker, who is the creator of the innovative “apartments for like” [38] structures in the Netherlands, such as the search for the happiness of the residents as a compass, or “be the boss of your own life”, which is close to “let it decide”, or “use it or lose it, which is” close to “let it happen” and “stimulate”. In addition to the demands and resources of the work, it seems important to pay attention to a third essential aspect, which is the aspect of what the workers seek to achieve; it establishes beyond what the organization asks them in interaction with the people they accompany. This search leads to self-instrumentalization and satisfaction when the objectives are achieved.

### 4.2. The Protective and Constructive Processes of Controlling Self-Commitment and Resource Development

Facing the commitment of oneself through love and empathy and the daily confrontation with the idea of death, suffering, and the difficulties of fulfilling one’s role as a companion, a defense process is put in place, underpinning the individual and collective need to work on the dosage, modulation, and alternation between the commitment and disengagement of oneself, emotionally and cognitively from work situations. This delicate balance between engagement and distancing is expressed through the concepts of the practice of “barriers”, “handing over”, “letting go”, “stepping back”, and “good distance.” They suggest that there are limits of investment that should not be exceeded in the normal course of work. The literature shows that empathy can improve patient care in health care services [39] while at the same time being a precursor to burnout in personal services [40]. Thus, the challenge of the “right distance” [41,42] is huge. All the more so as the difficulties of measuring the commitment on the ground remain significant [43]. Furthermore, the recent work of Schaufeli et al. [44], developing a new tool for measuring and diagnosing individual burnout to overcome the limitations of the MBI of Masclah et al., has shown the centrality of emotional and cognitive disorders in burnout and their causal link with energy. This only underlines the importance of continuing to study and strengthen the protective processes revealed. The recent work of Lehto et al. also mentions these barriers [45], and on a related issue, a very interesting analysis of individual strategies of emotional and cognitive regulation has recently been carried out in this sense [46], which in a way recalls the work of Lazarus [47] on coping strategies in relation to stress. The latter lacks the consideration of interpersonal and organizational dynamics. In practice, our observations lead us to believe that training institutions and reception structures must better prepare professionals for this self-regulation and collective regulation, particularly in their initial training. However, this aspect cannot be sufficient, as the balance between demands and resources, behind which lies the possibility of fulfilling one’s role, appears to us to be an important determinant of the balance between proximity and distance achieved by professionals at individual and inter-individual levels.

### 4.3. Resources and their Management in French Public Assisted Living Residences

Unsurprisingly, we have verified the importance of the resource development perspective as the mainspring of the continuity of commitment in spite of difficult-to-achieve objectives. For the cases observed, amid organizational resources, one has access to specialized training, in particular the specialized training for Gerontological Care Assistants (GCA), which is reserved for care assistants and medical–psychological assistants [48]. In our opinion, the question arises regarding its opening toward other paramedical disciplines, emphasizing, as the analysis shows, that the improvement in the care of the resident/patient is underpinned by the transfer of knowledge acquired within the teams on a daily basis. The improvement of the information-sharing system is a point that also appears central to the development of resources in French public EHPADs.

If everyone in France agrees on the need to redesign the work of assisted living residences, this study allows us to draw up some avenues. 

(1) The first level of transformation would be the one that refers to the revision of the calculation of the actual workload, in order to create long-lasting working conditions over time. Nowadays, the system creates a significant time pressure with disastrous consequences for professionals and organizations alike through the creation of a spiral of loss of individual, interpersonal, and organizational resources. The spiral dynamics of resources are beginning to be observed and analyzed in a few studies [49,50]. This procedural attention to the interconnection of resources and their evolution over time seems essential to us. Moreover, in order to contribute to the development of knowledge on this major focus of occupational health, a typology of the resources observed in this study was drawn up based on the frameworks proposed by Brummeluis et al. [49] and Nielsen et al. [51]. We synthesized and formalized the constructive structural elements of the system and the different resources observed and analyzed into organizational, inter-individual, and individual variables in Figure 4. Then, we ordered them according to what had emerged through our data and the literature as structural, determinant, volatile, and determined. Arrows and coloring indicate that structural working conditions and the systems that organize them affect volatile [17] and yet essential reservoirs of resources (individual, inter-individual, and organizational dynamics) such as atmosphere, confidence, or self-esteem.

(2) The second level of transformation is the management of skills and empowerment for professionals, in order to participate in creating long-lasting and desirable working conditions. In other words, instead of the current silo system that confines individuals into statutory boxes, it is necessary to develop a progressive empowerment system in which the actually developed skills are taken into account and preserved. It is a question of getting out of the dynamics of “dead-end” jobs to actually allow the following: (a) from an individual point of view, to project oneself into statuses and activities perceived as material and symbolic improvements; (b) from an organizational point of view, to capitalize on the development of resources, of knowledge and skills (including those located in important areas) developed in the activity and investment in academic training. Today’s caregivers, whether they are TSA, HSA, NA, or CNA, arrive in service and care training at best without specific training in gerontology. The recent creation of the title of advanced practice nurses in gerontology and the title of a care assistant in gerontology [48] go nevertheless in this direction. This is also in line with the advocacy of Beck et al. [52] for the empowerment of nursing assistants carried out 20 years ago in the United States to improve the quality of care.

(3) The third level is working prescription systems, the way expectations and resources are embedded in the organizational structure of work. It is completed by the detection and problem resolution system, which supports the first system by regulating the observed shifts. Lehto et al. [45] draw similar conclusions following their observation in the Midwest of the United States on the importance of being able to intervene in the determinants of work for the quality of work life. According to our observations, the first system seems to be very hampered by the dispersion of decision-making centers, which makes it very difficult to react, and by the distance from the activity, where the opportunity for field expertise and its recognition is lost. The second system was mainly activated in our observations in the informal part of the organization, its structural dimension suffering from the same defects as the first. However, the informal dimension of the organization suffers from a high degree of instability due to the very instability of the teams, management, and leadership. It should be noted that these centralized and hierarchical structures are the opposite of one of the 14 forces of magnet hospital identified in the United States in the 1980s [53]. About 40 years after the first work on the identification of these forces, a research-intervention on a French university hospital [54] is currently underway with the objective of decentralizing power. This is a challenging objective to implement in a French hospital system that is still very centralized and hierarchical, in which some of the assisted living residences in the study are included and in which it is more difficult to transform by team initiatives.

### 4.4. Limitations of the Study

While the method of data collection and analysis allows for a holistic understanding of the determinants and processes of occupational health, it has limitations. Firstly, the inclusions do not include people on sick leave, which would usefully complement the data collected from those present in the workplace on understanding the spiral of resource loss and its effects. Secondly, the dual status of observer and participant for the working groups and meetings limited the collection of data, which are rich times for understanding work and health at work. This could be rectified by the inclusion of a dedicated observer for the research and the digital recording of working sessions. Thirdly, although many of its elements are confirmatory of existing models and theories, the theory produced is situated; it would be classified by Strauss as substantive theory [29], which is built on an observational environment, which deserves to be compared to analyses of the determinants and processes of occupational health in other environments to make these elements applicable to any workplace. That is, to transform it into a formal theory. In addition, always with the aim of consolidating the results, it would also be interesting to construct a statistical survey protocol with the aim of producing a confirmatory analysis of the results obtained and thus measure the weight of the different elements noted and their interactions.

## 5. Conclusions

The objective of this qualitative study was to achieve a unique understanding that would qualitatively shed light on the social processes at work in the occupational health of caregivers in the French public assisted living residence system. The results show how their professional social role is defined beyond the institutional prescription through interaction with residents receiving care and services. Their efforts put into the realization of this social role are all the more important, as the necessary resources are not provided by the organization and the concern of the beneficiaries is important for these professionals. As long as they exhaust their cognitive and physical energy resources to compensate for the lack of resources in their role, the recognition of their commitment does not follow, and they find themselves at risk of losing reflexive control of their commitment. In addition, the lack of possibilities to improve the resources of work situations and/or professionals leads to a decline in their psycho-affective resources. Caught up in negative feelings about what is happening in their environment, they are bordering on the paralyzing identity crisis.

Our first results and our project have real theoretical but also practical implications. To our knowledge, this is the first “research-intervention” study, in a controlled trial mode, assessing the impact of empowering work organization on assisted living residences (EHPAD) in France. Our first results were presented to the different departments of the participating hospitals and then to the HCWs for the implementation of a local prevention plan aiming at improving the quality of life at work. More ever, comparison of the trial results (qualitative/quantitative approach) would help to identify other resources/constraints. Beyond the results of the “theoretical research” presented here, our study protocol and these first results have been and will be presented to the regional (Agence Régionale de la Santé/ARS) and national (Caisse Nationale de Retraite des Agents des Collectivités Locales/CNRACL) authorities in order to optimize the resources of establishments that receive/care for the elderly. In order to implement a global prevention plan, we also choose to present these original results in hospital management/health care workers school institutes and post-graduate courses (university degrees) in Nantes, Pays de la Loire, France.

## Figures and Tables

**Figure 1 ijerph-18-07286-f001:**
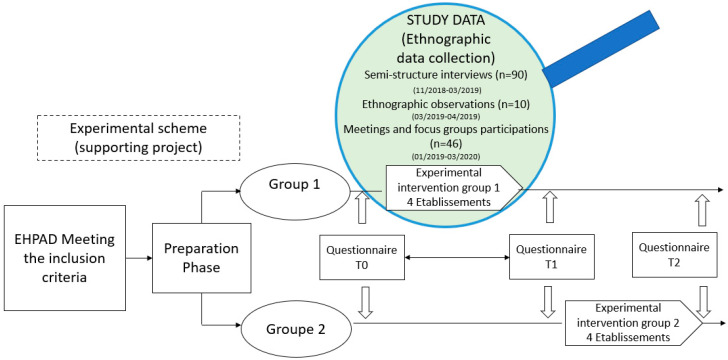
Flow collection of data of the study in the experimental scheme: “Empowerment and Quality of Life at Work in Retirement Homes” program.

**Figure 2 ijerph-18-07286-f002:**
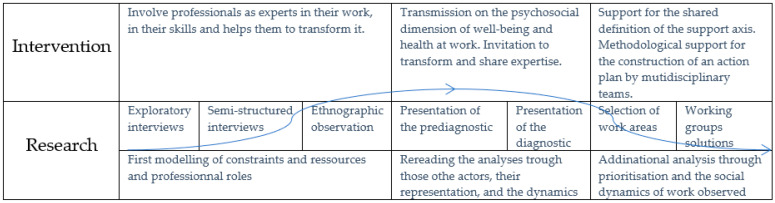
Dialectic of knowledge production between intervention and research.

**Figure 3 ijerph-18-07286-f003:**
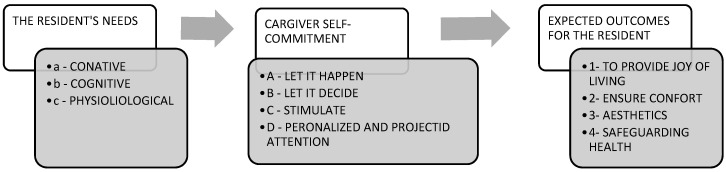
Concept map of health caregiver’s engagement and self-realization.

**Figure 4 ijerph-18-07286-f004:**
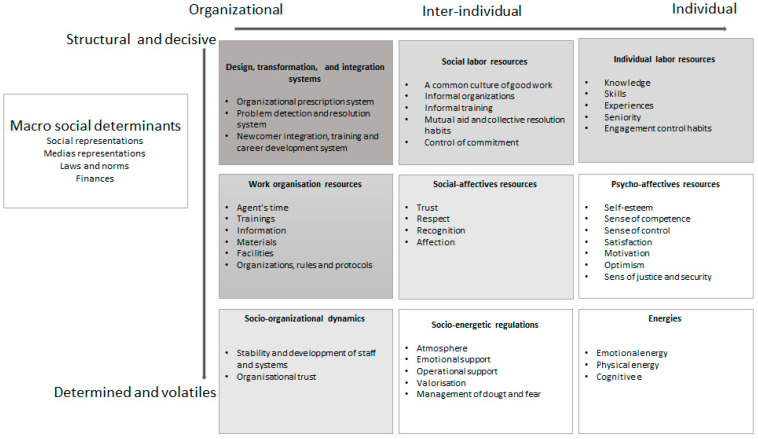
Summary of observed resources.

**Table 1 ijerph-18-07286-t001:** Description of the included EHPADs according to their workforce, geographical situation, and dependency of residents.

Establishments	Status	Work Force	Geograp-Hical Situation	Resident Breakdown According to Dependency			
				Help Needed for All (or Almost All) Activities	Help Needed Several Times a Day	Ponctual Help Needed	Total
a	Territorial Public Service	38	Metropolis	32	27	16	75
b	Hospital Public Service	29	Small town	28	20	1	49
c	Territorial Public Service	40	Metropolis	40	30	8	78
d	Hospital Public Service	136	Suburban countryside	130	54	5	189

**Table 2 ijerph-18-07286-t002:** Distribution of interviews classified by occupation.

	Realized	Transcribed
Health Care Aides (HCA)/(MPA)/Gerontological Care Assistants (GCAs)	41	16
Hospital Service Workers (HSW)/Territorial Service Workers (TSW)	23	16
State Registered Nurses (SN)	15	11
Health Care Managers and Administrative Managers	6	0
Others	5	0
Totals	90	43

**Table 3 ijerph-18-07286-t003:** Inclusions of interviews by the establishment.

Establishments	Interviews	Workforce	Participation Rate (%)
a	29	38	76.3
b	16	29	55.2
c	6	40	15.1
d	39	136	28.6
Total	90	243	37

**Table 4 ijerph-18-07286-t004:** The elements of the health care worker’s “own role”.

Dispositions	Love, patience, kindness, empathy
Objectives for residents	Wish and joy of living; comfort and cleanliness; aesthetics; physiological health
Guidelines	Letting it be; letting decide; to stimulate, to avoid the “slackness”, personalized attention
Axes of support	Conative, cognitive, physiological
Diachronic and relational dimension	Personalized and projected attention

## Data Availability

The data are not publicly available due to anonymity. The transcripts are available on request.

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
