# Peer review of "Psychosocial and Organizational Processes and Determinants of Health Care Workers’ (HCW) Health at Work in French Public EHPAD (Assisted Living Residences): A Qualitative Approach Using Grounded Theory"

_ijerph, 2021, doi:10.3390/ijerph18147286_

Round 1
Reviewer 1 Report
This is an interesting study that qualitatively demonstrated the processes and determinants at work in healthcare workers in the French public assisted living residence system.
(1) The authors performed interviews and observations in different periods. Is it possible to specify the time of interviews and observations in Figure 1 or a new figure so that readers could understand the study flow? In addition, the number of participants in each step should be listed. Otherwise, it is difficult to connect Figure 1, Table 1, and the text. It is also unclear what the 46 participations meant (Interviews? Observations? Participants?)
(2) All tables and figures: Tables and figures should be self-explanatory and can be understood independently. Abbreviations should be spelled out. For example, it is unclear the meaning of MPA/NA/GNA, HAS/TSA, and CNA in Table 1.
(3) Line 183: What does “AA” mean?
(4) Section 2.4 Analysis: It’s unclear whether the 16 interviews (line 191) come from the 16 transcribed interviews of MPA/NA/GNA (Table 1). A similar problem occurs in line 201, i.e., the 27 transcripts.
(5) Lines 225-226: “15,1%” should be “15.0%”? “76,3%” should be “76.3%”?
(6) Table 2: What is the meaning of 1, 5, 4, and 3? The number of establishments? Could the authors provide the definition?
(9) There are several grammatical and formatting problems. The manuscript should be English proofread carefully before being submitted to the second review.
Author Response
Dear reviewer 1, authors thank you very much for your comments which helped us to make the flow of our study clearer.
Please see the attached file for the responses.

Reviewer 2 Report
- This manuscript aims at understanding and explaining the processes and determinants at work that positively and negatively interfere with the professionals’ health in French public nursing home. For this purpose, 90 semi-structured interviews were recorded and 43 transcribed, 10 observations, and 46 25 participations in meetings and working groups were carried out in 4 public service and hospital 26 establishments.
I have some comments and suggestions that I describe below. Moreover, how were the participants selected?
- Firstly, I think the contribution of the research should be more explicitly stated.
- Please characterize the participants in a more detailed manner.
- Please provide additional information about the criteria you used to select the interviews.
- Did you used any software to analyze the data?
- The theoretical and practical implications of the research should be more developed and clearly stated.
- What are the limitations of your study?
- I suggest you take another pass through the manuscript to clean up grammar and usage issues.
Author Response
Dear reviewer 2,
Authors thank you very much for your comments which helped us to make the flow of our study clearer.
Please see the attached file for the responses.
Sincerly Yours.
